# Targeting Diet Quality at the Workplace: Influence on Cardiometabolic Risk

**DOI:** 10.3390/nu13072283

**Published:** 2021-06-30

**Authors:** Samira Amil, Isabelle Lemieux, Paul Poirier, Benoît Lamarche, Jean-Pierre Després, Natalie Alméras

**Affiliations:** 1Centre de recherche de l’Institut universitaire de cardiologie et de pneumologie de Québec—Université Laval, Québec, QC G1V 4G5, Canada; samira.amil@criucpq.ulaval.ca (S.A.); isabelle.lemieux@criucpq.ulaval.ca (I.L.); paul.poirier@criucpq.ulaval.ca (P.P.); Jean-Pierre.Despres@criucpq.ulaval.ca (J.-P.D.); 2École de nutrition, Faculté des sciences de l’agriculture et de l’alimentation, Université Laval, Québec, QC G1V 0A6, Canada; Benoit.Lamarche@fsaa.ulaval.ca; 3Faculty of Pharmacy, Université Laval, Québec, QC G1V 0A6, Canada; 4Centre Nutrition, santé et société (NUTRISS), Institut sur la nutrition et les aliments fonctionnels (INAF), Université Laval, Québec, QC G1V 0A6, Canada; 5Department of Kinesiology, Faculty of Medicine, Université Laval, Québec, QC G1V 0A6, Canada; 6VITAM—Centre de recherche en santé durable, CIUSSS de la Capitale-Nationale, Québec, QC G1J 0A4, Canada

**Keywords:** abdominal obesity, cardiorespiratory fitness, hypertriglyceridemic waist phenotype, lifestyle intervention, diet quality index

## Abstract

The American Heart Association criteria for cardiovascular health include overall diet quality (DQ). The present study evaluated the effect of a workplace health promotion program targeting DQ and physical activity on features of cardiometabolic risk (CMR). Before and after the 3-month intervention, 2260 employees (1462 men and 798 women) completed a health and fitness evaluation including assessment of DQ using a validated food-based questionnaire. After the 3-month lifestyle modification program, DQ increased significantly in both sexes (*p* < 0.0001) as well as physical activity level (*p* < 0.0001). A reduction in waist circumference (*p* < 0.0001) and improved lipid levels were also observed. Significant associations were found between changes in DQ index and changes in CMR variables in both men (standardized regression coefficients ranged from −0.19 (95% confidence interval: −0.26 to −0.12) to −0.29 (95% confidence interval: −0.34 to −0.25)) and women (standardized regression coefficients ranged from −0.18 (95% confidence interval: −0.25 to −0.11) to −0.27 (95% confidence interval: −0.41 to −0.13)). Multiple linear regression analyses showed a significant contribution of changes in the DQ index to the variation in some CMR variables, independent from changes in physical activity level and cardiorespiratory fitness. This study provides evidence that targeting DQ at the workplace is relevant to improve cardiometabolic health.

## 1. Introduction

According to the World Health Organization, 41 million deaths annually are attributed to noncommunicable diseases, representing 71% of worldwide mortality [1]. Cardiovascular diseases (CVD) are the first cause of deaths accounting for 31% of global deaths [2]. This proportion increases to 37% in people under the age of 70. Thus, the prevalence of chronic diseases remains high despite advances in medical treatments and procedures as well as in efforts invested in primary prevention. As a consequence, the societal and economic burden linked to these diseases represents a major public health issue which may be not sustainable by health systems in the future. Such a situation emphasizes the relevance of developing upstream prevention strategies to slow the CVD tide and its associated socioeconomic consequences. 

In order to address this issue, the American Heart Association (AHA) proposed in 2010 to move the focus from fighting CVD to promoting cardiovascular health and introduced the concept of ideal cardiovascular health as an approach to reduce the burden of CVD [3]. The AHA recommended to target seven heart-healthy metrics (Life’s Simple 7), four of which are lifestyle-based (not smoking, increasing physical activity, having a normal body mass index (BMI), eating a healthy diet), while the three others are based on achieving ideal clinical and laboratory measures (having normal blood cholesterol and glucose levels and normal blood pressure). Maintaining these metrics as close as possible to ideal levels has been shown to be associated with markedly low CVD incidence and mortality as well as with a low incidence of cancer [4,5,6]. Among the seven metrics of cardiovascular health, studies have reported that the least prevalent healthy behavior was a high diet quality (DQ), a criterion which was found to be met by less than 1% of individuals, making DQ the most deteriorated parameter among ideal cardiovascular health metrics [7,8].

Therefore, considering that only a very small proportion of the population adopts a high DQ, modifying this factor could have a major beneficial impact on cardiovascular health [9]. Although there is no consensus definition on how to assess DQ, several indices have been developed for assessing a population’s adherence to dietary patterns associated with cardiovascular health [10]. In this regard, DQ indices based on food items or food groups consumed rather than nutrients have been shown to provide good discrimination of CMR [11,12].

Among opportunities available, the workplace has been proposed as a relevant setting for implementing health promotion strategies [13,14]. According to 2019 statistics, the labor force participation rate in Canada was 65.8% [15], workers spending 6 to 9 h a day on average at their workplace. Thus, the environment provided by public/private employers has the potential to play a key role in the adoption of behaviors compatible with either maintenance of health or development of chronic diseases. In 2014, the World Health Organization proposed the workplace as a priority setting for health promotion [16]. To our knowledge, there is a paucity of published intervention studies conducted at the workplace that have targeted the adoption of healthy lifestyle habits including food-based overall DQ. The present study was therefore conducted to evaluate the relevance of assessing and targeting food-based overall DQ in the context of a workplace health promotion program and its effects on various indices of cardiometabolic health.

## 2. Materials and Methods

### 2.1. Participants

Our cohort is a convenience sample of employees involved in the “Grand Défi Entreprise” (GDE) project, a workplace health and wellness program which provided comprehensive cardiometabolic and cardiorespiratory health evaluations using a mobile risk assessment unit. Once the baseline evaluation is completed, the GDE also involves a 3-month lifestyle intervention where participating employees are asked to increase their physical activity level (PAL), improve their eating habits (DQ), and stop smoking [17,18].

All participants were volunteers. The intervention program took place between 2011 and 2019 and involved 28 participating organizations of the Province of Québec. No inclusion or exclusion criteria were used. This paper compares the CMR data at baseline and after the 3-month lifestyle intervention program obtained on a sample of 2260 workers (1462 men and 798 women) derived from an initial cohort of 5122 workers. Therefore, participants who did not participate to the intervention were excluded as well as participants with missing DQ data either at baseline or at the 3-month evaluation (see Figure 1 for Flowchart).

All measurements were performed in a single visit at baseline and at 3 months by trained healthcare professionals. All participants completed standardized questionnaires on medical history, current medication, and lifestyle behaviors (DQ, PAL, and smoking status). Data included anthropometric variables, body composition, waist circumference, lipid profile, and cardiorespiratory fitness (CRF). The local Institutional Review Board approved the study (20636), and participants provided their informed consent.

### 2.2. Assessment of Overall Diet Quality

The dietary screening tool (DST) [19] was selected to assess DQ. This questionnaire has been found helpful to provide personalized food-based nutritional recommendations to workers. Further details on this tool have already been published [20]. Compared to other DQ indices, this tool does not evaluate adherence to nutrition guidelines or dietary patterns [21,22]. The DST is based on the frequency of consumption of several food items or food groups (vegetables and fruits, grain products, dairy products, meat, and substitutes) as well as on the assessment of certain dietary habits such as the addition of sugar and fat, the consumption of alcohol, or the use of nutritional supplements. This questionnaire consists of 25 food- and behavior-specific questions associated with dietary habits and generates a DQ score, which varies from 0 (low DQ) to 100 (high DQ) [19]. The DST is therefore useful to identify, in 10 min, individuals at high nutritional risk which is one of its strengths.

### 2.3. Physical Activity Level

Reported PAL was measured at baseline using a self-administered, validated questionnaire that assesses leisure-time aerobic physical activity (cycling, walking, running, swimming, etc.) for each season during the year preceding the intervention [23]. During the 3-month intervention, using an electronic journal, workers compiled each aerobic period of 15 min of physical activity in their leisure time for the assessment of the cumulative physical activity time during a week. An average number of total minutes per week was subsequently calculated.

### 2.4. Anthropometric Measurements and Body Composition

Height and weight were obtained both at baseline and post-intervention according to standardized procedures, and BMI was calculated [24]. Waist circumference was assessed using standardized procedures [25]. Body composition (fat mass and body fat) was estimated with the Tanita body composition analyzer TBF-300A (Tanita Corporation, Arlington Heights, IL, USA) for the employees from organizations evaluated between 2011 and 2017. From 2018, the InBody 570 body composition analyzer was used (InBody Co., Seoul, Korea).

### 2.5. Lipid Profile

Nonfasting blood samples obtained from the forearm vein were collected into lithium heparin tubes and analyzed with an Abaxis Piccolo Xpress Chemistry Analyzer (Union City, CA, USA) to assess cholesterol fractions and triglyceride (TG) concentrations.

### 2.6. Cardiorespiratory Fitness

As already published, a submaximal treadmill exercise test was performed to assess CRF [17,26]. Maximal oxygen consumption (VO_2_max) was estimated by linear extrapolation [27] to age estimated maximal heart rate (220-age) [28] using ACSM’s Metabolic Equations and the least square method [29]. Two parameters were used as CRF endpoints: the heart rate at the standardized submaximal exercise workload (3.5 mph, 2% slope) and the estimated VO_2_max.

### 2.7. Hypertriglyceridemic Waist Phenotype

The hypertriglyceridemic waist (hyperTG waist) phenotype was used to identify individuals with visceral obesity and at risk for cardiometabolic abnormalities [30,31]. Criteria used were waist circumference ≥90.0 cm and TG ≥ 2.0 mmol/L for men or waist circumference ≥85.0 cm and TG ≥ 1.5 mmol/L for women [30,31].

### 2.8. Lifestyle Intervention

Details on the intervention have been published elsewhere [17]. After the baseline evaluation, workers were invited to form teams of five individuals to participate in an in-house competition. Throughout the 3-month intervention, physical activity and nutritional objectives were compiled on a web platform. Such ongoing collection of participants’ key behaviors made it possible to establish the team’s ranking in real time. The platform served as a lifestyle journal as well as a means to share and communicate progress within the team and at the workplace. Workers also received tips and recommendations. This friendly competition favored peer support helping individual changes in lifestyle habits. At the end of the intervention, prize incentives were offered by the management of participating organizations.

### 2.9. Statistical Analyses

Sex differences in baseline characteristics were tested by an unpaired t-test. A repeated measures analysis of variance was used to examine changes in variables between baseline and the 3-month follow-up. The normality assumption was verified with the Shapiro–Wilk test on residuals from the statistical model. The Brown and Forsythe’s variation of Levene’s test statistic was used to verify the homogeneity of variances. For most of the variables, these assumptions were not fulfilled. A repeated measures analysis of variance on ranks was therefore performed using the approach proposed by Brunner et al. [32]. As a significant sex*time interaction term was found for many cardiometabolic variables, analyses have been performed by sex. The McNemar’s test for paired data was performed to analyze changes in the proportion of hyperTG waist carriers. Pearson’s correlations were computed to measure the association between changes in the DQ index or PAL and changes in CMR variables. To investigate the relationship between changes in DQ or changes in PAL with changes in CMR risk variables, multiple linear regression models were performed. All statistical regression models were adjusted for medication use for lipids, hypertension, and diabetes as well as menopausal status in women. A second model including waist circumference was also performed, and the effect of potential confounding variables such as medication use for lipids, hypertension, and diabetes as well as menopausal status (in women) was further examined. Lastly, the potential contribution of interaction terms among studied variables was also assessed in regression analyses. Models with the lowest Akaike information criterion (AIC) were chosen. Finally, a one-way analysis of variance adjusted for baseline DQ index was performed to compare changes in CMR variables between quartiles of changes in the DQ index with Tukey–Kramer’s post hoc corrections for multiple comparisons. A *p* value ≤ 0.05 was considered as statistically significant. All statistical analyses were performed using SAS statistical package version 9.4 (SAS Institute, Cary, NC, USA).

## 3. Results

The mean age of the 2260 participants was 44.3 ± 10.1 years in men and 42.4 ± 10.6 years in women (range: 19 to 76 years of age), and 64.7% of participants were men. Almost half of participants were blue-collar workers (46.8%). Additional sociodemographic characteristics are presented in Table 1. At baseline, 40.5% of employees were overweight and 25.6% met the criteria for obesity (BMI ≥ 30 kg/m^2^). The prevalence of pre-existing treated diabetes and hypertension was 2.7% and 11.1%, respectively, and 10.2% of participants self-reported a history of dyslipidemia. Moreover, 13.7% of workers were active smokers or former smokers <12 months, and 56.9% of participants had never smoked.

Table 2 presents anthropometric, body composition, and lifestyle variables as well as CRF before and after the 3-month lifestyle modification program. A significant improvement of 7 units in the DQ index was observed in both sexes (*p* < 0.0001). At the end of the intervention, participants became significantly more active as revealed by the increase in PAL (*p* < 0.0001). Accordingly, the estimated VO_2_max also improved significantly in men (*p* < 0.0001), whereas the increase was of borderline significance in women (*p* = 0.055). As an additional CRF indicator, heart rate at a standardized exercise workload decreased significantly in both sexes (*p* < 0.0001 and *p* < 0.05 in men and women, respectively). These changes were accompanied by significant reductions in BMI, waist circumference, fat mass, and percent body fat in both sexes (*p* < 0.0001).

The plasma lipid profile also improved significantly after the 3-month lifestyle modification program (Table 3). For instance, total cholesterol, TG, and non-high-density lipoprotein (non-HDL) cholesterol concentrations declined significantly in both men (*p* < 0.0001) and women (*p* < 0.05). Significant decreases in low-density lipoprotein (LDL) cholesterol levels and the cholesterol/HDL cholesterol ratio were also observed in men (*p* < 0.0001). HDL cholesterol increased in men (*p* < 0.0001), whereas it decreased in women (*p* < 0.001) (Table 3).

Figure 2 presents changes in the proportion of hyperTG waist [30,31] in response to the workplace intervention program. Although the proportion of the hyperTG waist phenotype decreased in both sexes, the reduction only reached significance in men (*p* < 0.0001). In addition, relative changes in TG levels were significantly correlated with relative changes in waist circumference (r = 0.29, *p* < 0.0001, for both men and women).

Several significant negative relationships were observed between changes in the DQ index or PAL and changes in CMR markers (Figure 3). For instance, increases in the DQ index and PAL were associated with decreases in BMI, waist circumference, total cholesterol, non-HDL cholesterol, and TG. To examine the relationship between changes in the DQ index and changes in the CMR profile further, quartiles of changes in the DQ index were compared (Figure 4). This analysis revealed that men in the top quartile of change in the DQ index were also those who showed the most substantial reduction in waist circumference (*p* < 0.0001). They were also characterized by greater reductions in TG and non-HDL cholesterol levels compared to men in the first two quartiles (*p* < 0.05). In women, only changes in TG were significantly different across quartiles of changes in the DQ index (*p* < 0.05).

To quantify the independent associations of changes in the DQ index, PAL, and submaximal exercise heart rate to the 3-month variation in CMR markers, multiple linear regression analyses were performed (Table 4). In men, changes in anthropometric measures and body composition variables were mainly associated with the change in the DQ index (*p* < 0.0001) with smaller but significant associations of changes in PAL and exercise heart rate. On the other hand, in women, changes in PAL significantly contributed to explain changes in body composition indices, while changes in waist circumference and BMI were more associated with changes in the DQ index. Moreover, in men, changes in the lipid profile were explained by changes in the DQ index, PAL, and exercise heart rate. However, in women, changes in PAL contributed the most to overall changes in blood lipids (*p* < 0.05) except for changes in TG levels which were only associated with changes in the DQ index (*p* < 0.001). Finally, in order to examine to what extent the beneficial effect of DQ and PAL on CMR factors could be confounded by concurrent changes in waist circumference, this variable was added to the linear regression analysis (Table 5). In men, adjusting for waist circumference largely attenuated the association between changes in DQ and lipid variables. In women, changes in DQ and in PAL remained significantly associated with changes in several lipid variables even after adjustment for concurrent changes in waist circumference.

Additional statistical analyses were performed in order to control for menopausal status and medication use (lipids, hypertension, diabetes). After taking into account these potential confounding variables, similar findings were observed (Appendix A). Lastly, the potential contribution of interaction terms among studied variables was assessed. In women, results did not change following the addition of interaction terms into multiple linear regression analyses. In men, interaction terms only significantly contributed to changes in the cholesterol/HDL cholesterol ratio and changes in TG levels. The interaction between changes in PAL and changes in waist circumference (R^2^: 1.7%, *p* < 0.001) as well as changes in waist circumference (R^2^: 8.4%, *p* < 0.0001) and changes in exercise heart rate (R^2^: 0.5%, *p* ≤ 0.05) contributed to explain changes in the cholesterol/HDL cholesterol ratio. In addition, the interaction between changes in exercise heart rate and changes in waist circumference (R^2^: 1.2%, *p* < 0.05) contributed to explain changes in TG levels along with changes in waist circumference (R^2^: 9.4%, *p* < 0.0001), changes in PAL (R^2^: 0.8%, *p* < 0.05), and changes in DQ (R^2^: 0.5%, *p* ≤ 0.05).

## 4. Discussion

In the present study, our objectives were to evaluate whether: 1—DQ could be improved by our workplace lifestyle modification program; and 2—changes in DQ would have a favorable influence on features of cardiometabolic health. Results of the present study provide evidence that food markers of overall DQ can be assessed and targeted at the workplace in order to improve cardiometabolic health substantially. For instance, a large proportion of participants (76%) improved their DQ index (by at least 1 unit) after the 3-month intervention program. Among these participants, the median increase in DQ index was 10 (interquartile range: 5, 16) in men and was 9 (interquartile range: 5, 16) units in women. In a previous cross-sectional analysis conducted on our cohort of male and female workers that used the same DQ assessment tool, we had previously reported marked differences in features of CMR when employees were classified on the basis of their DQ index [20].

Results of this intervention show that DQ can be assessed and targeted at the workplace. Indeed, in our study, changes in a simple food-based index were accompanied by significant reductions in waist circumference and by improved lipid levels. To our knowledge, there are only a few published intervention studies targeting lifestyle changes at the workplace that included a large number of employees which assessed a comprehensive CMR profile (e.g., body fat distribution, lipid levels, submaximal treadmill exercise test). In a workplace-based lifestyle intervention program conducted in a sample of employees at high risk of CVD and involving a team of health professionals (nurse practitioner, registered dietician, exercise physiologist, certified diabetic educator, and registered nurse), Rouseff et al. [33] reported significant improvements in several CMR markers such as percent body fat, blood pressure, lipid parameters, as well as CRF. One novel aspect of our study is our finding that it appears possible to target overall DQ with a food-based approach instead of using traditional nutritional interventions focused on dietary fat reduction and/or caloric restriction in order to have beneficial impacts on cardiometabolic health. Such an approach is in line with recent recommendations of nutrition experts [9,11,34,35]. As previously reported [20], we found significant differences in DQ, men having a poorer DQ index than women at baseline (mean ± SD: 60.5 ± 12.3 and 64.2 ± 11.6, for men and women, respectively, *p* < 0.0001). However, both sexes significantly improved their DQ in response to the program (by 7 units). Additionally, men of the present study were characterized by a more deteriorated baseline CMR profile than women (data not shown), although significant changes were observed in both sexes. In this regard, greater improvements in CMR markers are expected in men since lifestyle interventions have a greater effect on CVD risk factors in higher-risk populations [17,33,36,37,38]. In addition, in both men and women, changes in DQ were found to be associated with changes in adiposity and lipid levels (Figure 3). These findings show that the food-based index of DQ could discriminate CMR. In addition, our findings indicate that being successful in implementing meaningful changes in DQ as measured by this index predicted significant changes in CMR variables. Targeting DQ with a brief and food-based strategy has been effectively shown to improve DQ and reported to be as effective as other lifestyle interventions in improving CMR markers [39,40,41,42,43,44].

One unique feature of this workplace intervention aiming at improving DQ is that we also simultaneously considered PAL and CRF as important potential confounders of cardiometabolic health in our statistical analyses. Of course, one cannot exclude the possibility that employees who showed the most substantial improvements in their DQ were also those who became the most physically active. In this regard, it has been reported that the most active individuals are also those characterized by a better DQ [45]. In line with this possibility, we found a weak but significant association between changes in the DQ index and changes in PAL (r = 0.22, *p* < 0.0001 and r = 0.27, *p* < 0.0001 in men and women, respectively). In the present study, we found that whereas CRF assessed by estimated VO_2_max was significantly increased in men, the change in VO_2_max was only of borderline significance in women. However, submaximal HR was decreased in both men and women, a finding clearly showing that the change in PAL induced an improvement in submaximal working capacity in both sexes, while not being enough to improve maximal oxygen consumption in women. These results suggest that women may have responded to the lifestyle intervention by improving further their DQ and by doing more physical activity of moderate intensity, whereas men may have performed more vigorous physical activity. Although we did not assess physical activity intensity in the present study, it has been reported that VO_2_max is more likely to be improved with higher-intensity endurance exercise [46,47], while low-intensity exercise may not always be sufficient to improve CRF [48,49]. This sex difference in how workers adopted different strategies to improve their lifestyle habits will require further investigation.

Therefore, changes in PAL observed at the end of the 3-month intervention program may have also contributed to explain the beneficial changes found in CMR markers. This notion is supported by findings showing that across quartiles of DQ index changes, a reduction in waist circumference was observed (Figure 4). Moreover, in both sexes, changes in PAL were also associated with changes in features of cardiometabolic health (Figure 3). Thus, one novel aspect of our workplace intervention was to quantify the respective contributions of changes in the DQ index vs. changes in PAL to the improvement of cardiometabolic health. To examine this issue, we performed multiple linear regression analyses which revealed that in men, changes in the DQ index were related to changes in most features of CMR, with a modest but significant additional contribution of changes in PAL and CRF (Table 4). In women, whereas changes in the DQ index were independently associated with changes in CMR variables, changes in PAL had a major influence on the variation in adiposity variables, with no independent influence of CRF. While we propose that such potential sex differences should deserve further attention, we put forward the hypothesis that since women had a better DQ and were less physically active than men at baseline, their increase in physical activity, despite not being performed at the intensity required to improve maximal oxygen consumption (estimated by the VO_2_max), it could have nevertheless played a role in inducing weight loss and loss of abdominal fat (waist circumference). In men, we propose that since their DQ was lower than women at baseline, improving their DQ and performing more vigorous physical activity which translated into improved CRF could be key contributing factors to explain the beneficial effects of the program on their cardiometabolic health.

Finally, as both men and women reduced their waist circumference in response to the program, a key remaining question was to test whether improving DQ had an influence on CMR after controlling for concomitant changes in waist circumference. Our findings are concordant with the notion that changes in waist circumference in men largely explained the influence of changes in DQ on lipid variables, whereas in women changes in DQ and PAL remained associated with changes in several lipid variables after controlling for changes in waist circumference (Table 5). These results suggest that while targeting food-based DQ is relevant to improve cardiometabolic health, the concomitant effect of such an intervention (diet and physical activity) on waist circumference is an important contributor to the association between DQ and cardiometabolic health. As already highlighted in a consensus paper [50], the reduction in waist circumference through lifestyle changes is likely to generate a decrease in morbidity and mortality risk by improving intermediate CMR markers. One of the reasons explaining this beneficial association would be the mobilization of the high-risk visceral fat generated by the lifestyle modification program [51]. Although not measured, we can speculate that employees of the present study for whom a reduction in waist circumference was observed were also characterized by a reduction in visceral fat as suggested by the decrease in carriers of the hyperTG waist phenotype, a clinical marker of visceral obesity [30,31].

A strength of our workplace lifestyle intervention program is the collection of a comprehensive CMR profile including the evaluation of CRF by a submaximal exercise treadmill test. In addition, our study sample was heterogeneous in terms of education level, socio-economic status, health profile, and job characteristics and demands. In order to make sure that we could evaluate the effects of our program on a population as heterogeneous as possible in terms of baseline characteristics, we had no criteria for employees’ inclusion or exclusion. Furthermore, this study is a unique “real world intervention”, as it took place in different and uncontrolled work environments. The DST allows us to assess overall DQ without any intermediate method (24 h recall, quantitative food frequency questionnaire, etc.).

However, our study has some limitations. Firstly, despite the fact that we did not have inclusion/exclusion criteria, it is obvious that our sample of workers is not representative of the entire workforce as all participants were volunteers, an obvious selection bias. These participants are most likely to be more health-conscious and consequently may had responded to the program to a greater extent compared to the general population. Moreover, although we used a validated questionnaire, PAL is notoriously over-reported when compared to direct measurements [52] at baseline. Thereby, the change in PAL may be underestimated by an overestimation of baseline PAL. Thus, its independent contribution to variations in the CMR profile may have been underestimated in the regression model. Secondly, as our intervention is short term (3 months), there is clearly a need to document the long-term impact of that type of program. As a consequence, we cannot speculate about the long-term effect and adherence to our intervention.

## 5. Conclusions

In summary, this pragmatic 3-month lifestyle intervention provides evidence that a key behavior influencing cardiometabolic health such as DQ can be assessed and targeted at the workplace using a simple food-based questionnaire. Furthermore, the contribution of changes in DQ to the improvement in CMR remained significant after controlling for changes in PAL. Improving DQ through lifestyle interventions in the workplace may have direct effects on CMR, particularly in women, as well as indirect effects on both men and women through effects on abdominal adiposity. Finally, findings of the present study support the proposal from a recent consensus group that waist circumference is a simple and useful marker of the influence of healthy/unhealthy behaviors [50].

## Figures and Tables

**Figure 1 nutrients-13-02283-f001:**
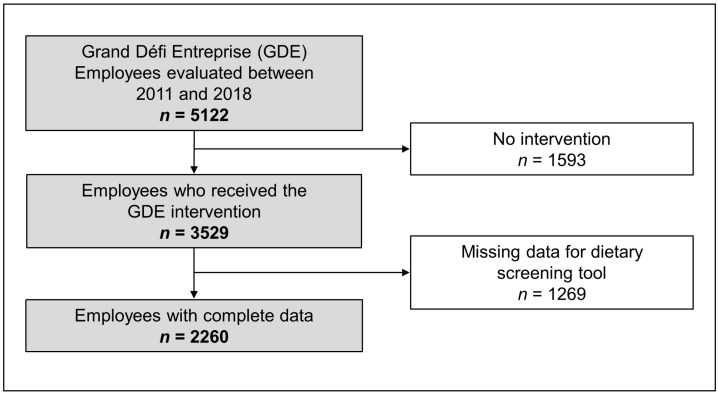
Flowchart of participants and reasons for exclusion for the present analyses.

**Figure 2 nutrients-13-02283-f002:**
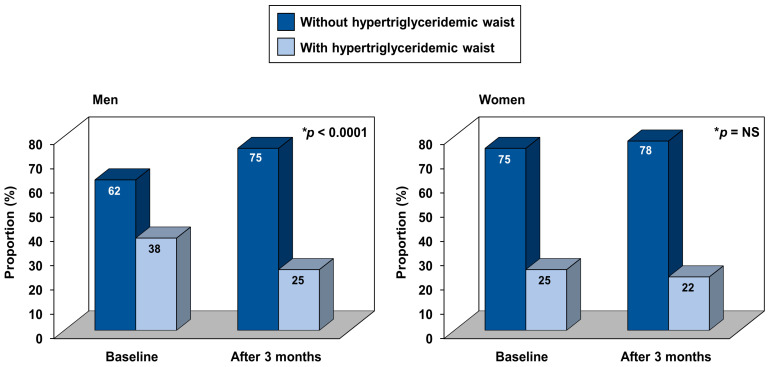
Changes in carriers of the hypertriglyceridemic waist phenotype in response to the 3-month lifestyle modification program. * The statistical difference has been quantified according to the McNemar’s test.

**Figure 3 nutrients-13-02283-f003:**
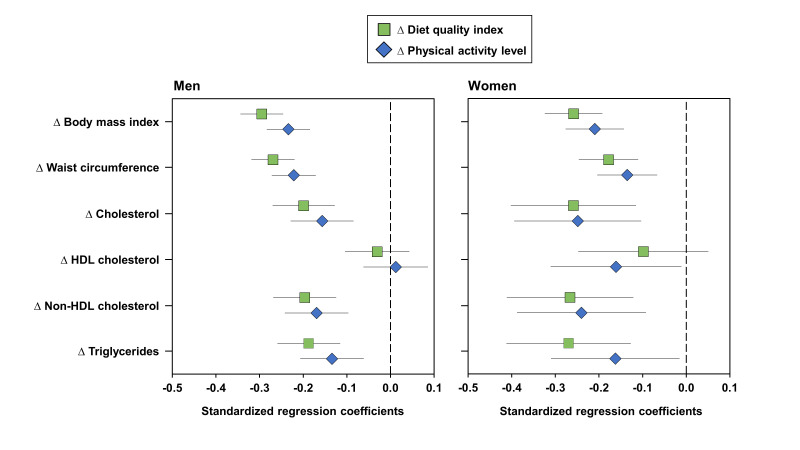
Standardized regression coefficients (with 95% confidence intervals) for the association between changes in diet quality index or physical activity level and changes in cardiometabolic risk markers in men and women. Models are adjusted for medication use (lipids, hypertension, and diabetes) and menopausal status in women. HDL: high-density lipoprotein; NHDL: non-HDL cholesterol.

**Figure 4 nutrients-13-02283-f004:**
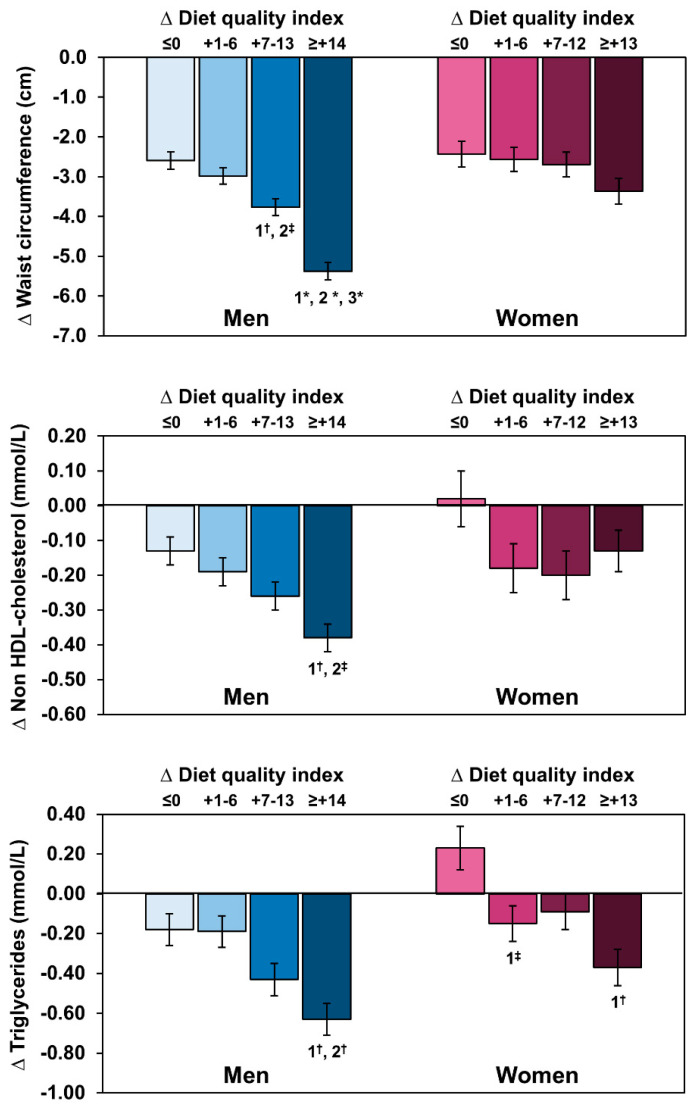
Changes in waist circumference, non-HDL cholesterol, and triglycerides according to quartiles of changes in diet quality index in response to the 3-month lifestyle modification program. Data represent least-squares means ± standard error to the mean adjusted for the baseline diet quality index. ^‡^
*p* < 0.05; ^†^ *p* < 0.001; * *p* < 0.0001.

**Table 1 nutrients-13-02283-t001:** Employees’ sociodemographic characteristics.

Variables	%
**Sex**	
Male	64.7
Female	35.3
**Ethnicity**	
Caucasian	94.3
Afro-Canadian	1.3
Latino-Canadian	0.9
Asian	0.4
First Nations	0.3
Others	2.9
**Marital status**	
Married/cohabiting	74.8
Unmarried	18.1
Separated/divorced	6.6
Widow/widower	0.5
**Employee categories**	
White collar	39.7
Blue collar	46.8
Unknown	13.5
**Household income**	
Low (<CAN 50,000)	26.6
Medium (CAN 50,000–80,000)	14.1
High (>CAN 80,000)	54.2
Unknown	5.2
**Education**	
<High school	3.9
High school	36.9
College	29.3
University	19.9
Post-graduate	10.0

**Table 2 nutrients-13-02283-t002:** Employees’ characteristics as well as markers of lifestyle habits before and after the 3-month lifestyle modification program.

	Baseline	3 Months	Δ
**Men**			
**Anthropometric measurements** **and body composition (*n*)**	1430–1462	1459–1462	1429–1462
Body mass index (kg/m^2^)	27.4 (24.9, 30.4)	26.8 (24.3, 29.4)	−0.5 (−1.1, 0.0) *
Waist circumference (cm)	97.0 (89.3, 104.9)	92.9 (85.9, 100.6)	−3.4 (−5.8, −1.2) *
Fat mass (kg)	20.4 (15.3, 26.8)	18.6 (14.1, 24.5)	−1.3 (−3.1, −0.1) *
Body fat (%)	24.4 (20.1, 28.7)	22.8 (18.9, 26.8)	−1.2 (−2.8, 0.0) *
**Lifestyle habits (*n*)**	1462	1452–1462	1452–1462
Diet quality index	61 (52, 70)	69 (60, 76)	7 (1, 14) *
Physical activity level (min/week)	210 (84, 378)	222 (131, 339)	18 (−108, 121) *
**Cardiorespiratory profile (*n*)**	1384–1441	684–708	669–702
Exercise heart rate (bpm)	112 (103, 121)	108 (100, 115)	−4 (−10, 2) *
Estimated VO_2_max (mL/min/kg)	41.5 (35.5, 48.3)	43.2 (36.7, 51.5)	1.9 (−3.0, 7.2) *
**Women**			
**Anthropometric measurements** **and body composition (*n*)**	793–798	798	793–798
Body mass index (kg/m^2^)	25.0 (22.1, 28.9)	24.5 (21.7, 28.4)	−0.4 (−0.9, 0.1) *
Waist circumference (cm)	85.3 (77.1, 96.8)	82.8 (75.2, 93.4)	−2.5 (−5.1, −0.1) *
Fat mass (kg)	21.7 (15.9, 30.0)	20.8 (14.6, 28.3)	−1.1 (−2.5, 0.1) *
Body fat (%)	33.0 (27.2, 39.5)	31.7 (25.7, 38.1)	−1.2 (−2.5, 0.1) *
**Lifestyle habits (*n*)**	798	798	798
Diet quality index	65 (57, 72)	72 (65, 79)	7 (1, 13) *
Physical activity level (min/week)	168 (84, 294)	211 (132, 314)	37 (−61, 126) *
**Cardiorespiratory profile (*n*)**	725–776	170–176	164–173
Exercise heart rate (bpm)	125 (114, 135)	120 (109, 131)	−3 (−9, 5) ^‡^
Estimated VO*_2_*max (mL/min/kg)	33.0 (27.7, 39.8)	34.1 (28.9, 40.5)	1.0 (−2.7, 5.1) ^§^

Data represent median and interquartile range (25th and 75th percentile). *n*: range of participants. ^‡^
*p* < 0.05; * *p* < 0.0001; ^§^
*p* = 0.055.

**Table 3 nutrients-13-02283-t003:** Plasma lipid profile before and after the 3-month lifestyle modification program.

	Baseline	3 Months	Δ
**Men**			
**Lipid variables (*n*)**	1418–1462	721–728	712–728
Total cholesterol (mmol/L)	4.70 (4.10, 5.30)	4.40 (3.90, 5.00)	−0.20 (−0.50, 0.10) *
LDL cholesterol (mmol/L)	2.50 (2.00, 3.00)	2.40 (2.00, 2.80)	−0.10 (−0.40, 0.20) *
HDL cholesterol (mmol/L)	1.25 (1.07, 1.47)	1.25 (1.07, 1.45)	0.03 (−0.07, 0.13) *
Non-HDL cholesterol (mmol/L)	3.40 (2.85, 3.95)	3.16 (2.64, 3.65)	−0.20 (−0.50, 0.09) *
Cholesterol/HDL cholesterol	3.70 (3.10, 4.30)	3.50 (3.00, 4.20)	−0.20 (−0.56, 0.10) *
Triglycerides (mmol/L)	1.81 (1.23, 2.72)	1.44 (1.00, 2.26)	−0.26 (−0.85, 0.22) *
**Women**			
**Lipid variables (*n*)**	780–794	180–182	179–182
Total cholesterol (mmol/L)	4.60 (4.10, 5.10)	4.40 (4.00, 5.10)	−0.10 (−0.40, 0.20) ^†^
LDL cholesterol (mmol/L)	2.40 (2.00, 2.80)	2.40 (2.00, 2.80)	0.00 (−0.30, 0.20) ^‡^
HDL cholesterol (mmol/L)	1.60 (1.34, 1.80)	1.44 (1.28, 1.68)	−0.02 (−0.15, 0.10) ^†^
Non-HDL cholesterol (mmol/L)	2.97 (2.48, 3.51)	2.94 (2.48, 3.41)	−0.12 (−0.34, 0.13) ^‡^
Cholesterol/HDL cholesterol	2.90 (2.50, 3.40)	3.10 (2.60, 3.50)	−0.10 (−0.30, 0.10)
Triglycerides (mmol/L)	1.18 (0.87, 1.69)	1.05 (0.79, 1.55)	−0.10 (−0.40, 0.14) *

Data represent median and interquartile range (25th and 75th percentile). HDL: high-density lipoprotein; LDL: low-density lipoprotein; *n*: range of participants. ^‡^
*p* < 0.05; ^†^ *p* < 0.001; * *p* < 0.0001.

**Table 4 nutrients-13-02283-t004:** Multiple linear regression models showing the independent associations of changes in diet quality index, physical activity level, and exercise heart rate on changes in cardiometabolic markers.

	Total R^2^ × 100	Partial R^2^ × 100 ∆ DQ Index	Partial R^2^ × 100 ∆ PAL	Partial R^2^ × 100 ∆ Exercise HR
**Men**				
**Anthropometric measurements** **and body composition (*n* = 669–697)**				
∆ Body mass index	15.7	10.5 *	1.5 ^†^	3.6 *
∆ Waist circumference	13.8	10.7 *	1.8 ^†^	1.3 ^‡^
∆ Fat mass	13.7	11.0 *	1.2 ^‡^	1.5 ^†^
∆ Body fat	9.6	8.3 *	0.9 ^‡^	0.5
**Lipid profile (*n* = 678–694)**				
∆ Total cholesterol	5.8	4.1 *	1.7 ^†^	–
∆ LDL cholesterol	0.4	0.4	–	–
∆ HDL cholesterol	1.8	0.4	–	1.5 ^‡^
∆ Non-HDL cholesterol	5.8	3.8 *	2.0 ^†^	–
∆ Cholesterol/HDL cholesterol	4.4	0.9 ^‡^	2.4 *	1.1 ^‡^
∆ Triglycerides	5.9	3.2 *	1.3 ^‡^	1.5 ^‡^
**Women**				
**Anthropometric measurements** **and body composition (*n* = 168–173**)				
∆ Body mass index	9.7	6.4 ^†^	3.3 ^‡^	–
∆ Waist circumference	9.5	5.9 ^‡^	3.5 ^‡^	–
∆ Fat mass	14.5	3.0 ^‡^	11.5 *	–
∆ Body fat	11.5	–	11.5 *	–
**Lipid profile (*n* = 170–173)**				
∆ Total cholesterol	14.0	4.6 ^‡^	9.4 *	–
∆ LDL cholesterol	2.4	–	2.4 ^‡^	–
∆ HDL cholesterol	4.3	–	4.3 ^‡^	–
∆ Non-HDL cholesterol	12.9	4.6 ^‡^	8.3 ^†^	–
∆ Cholesterol/HDL cholesterol	–	–	–	–
∆ Triglycerides	9.2	7.9 ^†^	1.3	–

DQ: diet quality; HDL: high-density lipoprotein; HR: heart rate; LDL: low-density lipoprotein; PAL: physical activity level; *n*: range of participants; – not included in the model due to lack of significance; ^‡^
*p* < 0.05; ^†^ *p* < 0.001; * *p* < 0.0001.

**Table 5 nutrients-13-02283-t005:** Multiple linear regression models showing the independent associations of changes in diet quality index, physical activity level, exercise heart rate, and waist circumference on changes in lipid variables.

	Total R^2^ × 100	Partial R^2^ × 100 ∆ DQ Index	Partial R^2^ × 100 ∆ PAL	Partial R^2^ × 100 ∆ Exercise HR	Partial R^2^ × 100 ∆ Waist
**Men**					
**Lipid profile (*n* = 678–694)**					
∆ Total cholesterol	9.8	1.4 ^‡^	1.0 ^‡^	–	7.3 *
∆ LDL cholesterol	0.8	–	–	0.3	0.5
∆ HDL cholesterol	1.8	0.4	–	1.5 ^‡^	–
∆ Non-HDL cholesterol	12.4	0.6 ^‡^	1.3 ^‡^	–	10.5 *
∆ Cholesterol/HDL cholesterol	9.9	–	1.0 ^‡^	0.5	8.4 *
∆ Triglycerides	11.4	0.5	0.8 ^‡^	0.8 ^‡^	9.4 *
**Women**					
**Lipid profile (*n* = 170–173)**					
∆ Total cholesterol	17.1	3.1 ^‡^	9.4 *	–	4.6 ^‡^
∆ LDL cholesterol	2.4	–	2.4 ^‡^	–	–
∆ HDL cholesterol	4.3	–	4.3 ^‡^	–	–
∆ Non-HDL cholesterol	16.1	3.0 ^‡^	8.3 ^†^	–	4.8 ^‡^
∆ Cholesterol/HDL cholesterol	–	–	–	–	–
∆ Triglycerides	13.4	4.6 ^‡^	–	–	8.9 *

DQ: diet quality; HDL: high-density lipoprotein; HR: heart rate; LDL: low-density lipoprotein; PAL: physical activity level; *n*: range of participants; – not included in the model due to lack of significance; ^‡^
*p* < 0.05; ^†^ *p* < 0.001; * *p* < 0.0001

## Data Availability

The data presented are available on request from the corresponding author.

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
