# Peer review of "Targeting Diet Quality at the Workplace: Influence on Cardiometabolic Risk"

_nutrients, 2021, doi:10.3390/nu13072283_

Round 1
Reviewer 1 Report
The manuscript nutrients-1259079 entitled " Targeting Diet Quality at the Workplace: Influence on Cardiometabolic Risk" is an original manuscript describing the effect of a workplace health promotion program targeting diet quality and physical activity on features of cardiometabolic risk. This is a well-structured and interesting manuscript. There are few points I would suggest to get clarified:
- Please include detailed information about dietary screening tool in the methodology session.
- Please include detailed information about lifestyle intervention in the methodology session.
- Woulditbepossibletoincludeatablewiththesociodemographicdataoftherespondents?

Reviewer 2 Report
In their paper “Targeting Diet Quality at the Workplace: Influence on Cardiometabolic Risk” Amil et al. evaluated the effect of a workplace health promotion program on diet quality and cardiometabolic risk in over 2200 subjects.
The aim of the study is clear and this is also true for the study design. In general, the paper is interesting and very well written, in particular, the introduction and discussion section. The data are also fine, however the last paragraph in the results section dealing with the linear regression could be improved. Apart from demonstrating only partial R^2 , authors may also focus on the possibility of interaction/mediation of the mentioned parameters. The results part would clearly benefit from elaborating these points and/or adding additional tests.
Apart from that, further potential confounders (menopause, drugs) were just mentioned, but data were "not shown”. This might be added.
